# Wide-range lifetime-tunable and responsive ultralong organic phosphorescent multi-host/guest system

Zongliang Xie [1], Xiayu Zhang[2], Hailan Wang[1], Cheng Huang[1], Haodong Sun[1], Mengyang Dong[1], Lei Ji[1], Zhongfu An [3], Tao Yu [1✉] & Wei Huang [1,3,4✉]

The rational lifetime-tuning strategy of ultralong organic phosphorescence is extraordinarily important but seldom reported. Herein, a series of multi-host/guest ultralong organic phosphorescence materials with dynamic lifetime-tuning properties were reported. By doping a non-room-temperature phosphorescence emitter into various solid host matrices with continuously reduced triplet energy levels, a wide-range lifetime (from 3.9 ms gradually to 376.9 ms) phosphorescence with unchangeable afterglow colors were realized. Further studies revealed that the host matrices were employed to afford rigid environment and proper energy levels to generate and stabilize the long-live triplet excitons. Meanwhile, these multi-host/guest ultralong organic phosphorescence materials also exhibited excitation-dependent phosphorescence and temperature-controlled afterglow on/off switching properties, according to the virtue of various photophysical and thermal properties of the host matrices. This work provides a guiding strategy to realize lifetime-tuning ultralong organic phosphorescence with lifetime-order encoding characteristic towards widespread applications in time-resolved information displaying, higher-level security protection, and dynamic multi-dimensional anti-counterfeiting.

[1] Frontiers Science Center for Flexible Electronics, Xi'an Institute of Flexible Electronics (IFE) & Xi'an Institute of Biomedical Materials and Engineering (IBME), Northwestern Polytechnical University, Xi'an, China. [2] School of Packaging and Materials Engineering, Hunan University of Technology, Zhuzhou, P. R. China. [3] Key Laboratory of Flexible Electronics (KLOFE) & Institute of Advanced Materials (IAM), Nanjing Tech University (NanjingTech), Nanjing, China. [4] Key Laboratory for Organic Electronics and Information Displays & Institute of Advanced Materials, Jiangsu National Synergistic Innovation Center for Advanced Materials, Nanjing University of Posts and Telecommunications, Nanjing, China. ✉email: iamtyu@nwpu.edu.cn; iamwhuang@nwpu.edu.cn

Ultralong organic phosphorescence (UOP) materials, also called the organic afterglow materials, have drawn great attention in the fields of organic light-emitting devices, anticounterfeiting, information encryption, flexible electronic devices and biosensing, owing to their superior performances to harvest triplet state energy and long-lived released processes[1–9]. Generally, high-efficiency UOP can be achieved by boosting the intrinsically spin-forbidden intersystem crossing (ISC) and suppressing the nonradiative relaxation pathways. Relatively, numerous UOP materials have been designed and developed by taking advantage of the design strategies such as crystallization engineering[10–12], H-aggregation[12,13], heavy atom effect[14], halogen bonding interactions[15,16], polymerization[17–21], and host/guest complexation[22–24]. Significant studies have been conducted to promote UOP lifetime, enhance the phosphorescence efficiencies and tune the afterglow emission colors[4,15,20–22]. Despite spectacular recent progress, lack of systematic research and exploration for lifetime-tunable properties of UOP materials inevitably hinders their development and practical applications.

Based on the characteristics of distinguishable time-resolved triplet-exciton decays, lifetime-tunable UOP materials have shown great potential applications in many fields. For instance, these materials can realize time-resolved information display and hide the encrypted information in a certain time node for higher-level security protection and dynamic anticounterfeiting. For a certain phosphor, the phosphorescence lifetime is determined by the rate constants of phosphorescence ($k_{phos.}$) and nonradiative deactivations ($k_{nr}$, including nonradiative decays generated from molecular motion and the triplet-exciton quenching caused by interactions with the surrounding substances such as oxygen and humidity)[6,25,26]. $k_{phos.}$ is related to the phosphorescent emitter and the $k_{nr}$ is also affected by the ambient condition. With fixed molecular structure (unchangeable emission color) and certain surrounding conditions, the phosphorescence lifetime is maintained. With regard to this, realizing wide-range lifetime-tunable UOP materials with certain phosphorescence color under certain conditions remains highly challenging.

Doping or embedding guest emitters into rigid matrices is a concise and effective strategy to obtain organic afterglow materials[7–9,22–24,27–36]. A rigid host matrix can efficiently inhibit motions of guest molecules and prevent triplet energy quenching from ambient humidity and oxygen[22–24,29]. Besides, proper triplet energy levels of the host matrices can also provide effective transition pathways for energy transfer between guest and host species, which is conducive to the generation and stabilization of triplet excitons[35] (Supplementary Fig. 1). By adjusting the energy gap ($\Delta E_{SGTH}$) between the lowest singlet state of the guest emitter ($S_{1G}$) and the lowest triplet state of host matrices ($T_{1H}$), the ISC process between them could be affected, wide-range lifetime-tunable multicomponent phosphorescence might be achieved (as shown in Fig. 1a). For this purpose, a series of rigid host matrices require proper triplet energy levels between the lowest singlet and triplet states of the guest emitter, along with the ability to isolate guest molecules and promote the energy transfer process. However, the effect of host matrices in doping systems remains unclear. Suitable host species are difficult to be rationally chosen for a certain guest emitter. These problems restrained the development of host/guest doping UOP materials.

Herein, we use 10-phenylphenothiazine (PzPh) as guest to construct a multi-host/guest UOP system (Fig. 1c). Besides high-efficiency phosphorescence, wide-range tunable lifetimes from 3.9 to 376.9 ms was successfully achieved by changing the host matrices as shown in Fig. 1b. Meanwhile, the phosphorescence color was kept at bright yellow with little-change CIE chromaticity coordinates within (0.39–0.42, 0.52–0.55). Triplet energy levels of the host species were demonstrated to play a vital role in the afterglow decay processes. Furthermore, these mH/G UOP materials also exhibited excitation-dependent UOP and temperature-controlled afterglow on/off switch properties. Simple flexible thin films were fabricated to show the lifetime-order encoding characteristics under different ambient conditions. Obviously, such a mH/G doping strategy can realize wide-range tunable lifetimes and show widespread applications in time-dependent information displaying, higher-level security protection, and dynamic multidimensional anticounterfeiting.

## Results

**Design and fabrication of the mH/G UOP system.** In this research, the eligible guest molecule requires relatively large singlet-triplet energy gap ($\Delta E_{ST}$) to accommodate the triplet energy platforms of various host matrices. Additionally, good compatibility between the host and guest species should also be considered. PzPh was chosen as the guest emitter for its suitable $\Delta E_{ST}$ (ca. 0.7 eV) and good compatibility. Pure PzPh has only a blue fluorescence band at 445 nm under UV excitation, no phosphorescence was observed at room temperature. It adopts a quasi-equatorial conformation with a butterfly-shape phenothiazine moiety folded in a dihedral angle range of 150.28°–162.44°, in which the phenyl ring is orthogonally connected to the phenothiazine[37–39] (Supplementary Figs. 2 and 3). With such a twisted conformation, PzPh is not easily embedded in a flat and compact host matrix. Therefore, we have carefully chosen a series of host matrices with suitable triplet energy levels between the lowest singlet and triplet states of PzPh and twisted conformations, namely benzophenone (BP), triphenylphosphine oxide (TPO), tetraphenyl-silane (TPSi), triphenylphosphine (TP), diphenyl-sulfone (SF), and triphenylamine (TPA). As shown in Supplementary Fig. 3, these matrices are closely connected in the crystalline state to afford rigid ambient condition, while the twisted molecular structure endows them with enough space to facilitate the insertion of the guest emitter. The large energy gap between the fluorescence and phosphorescence bands of PzPh can accommodate the phosphorescence spectra of host materials (Supplementary Fig. 4), revealing the possibility of phosphorescence lifetime tuning via the changeable energy gaps ($\Delta E_{SGTH}$) between host and guest species. These mH/G UOP materials can be manufactured by solution evaporation method, conventional melt-casting method or easy grinding method (details are shown in Methods), showing the advantages of concise and toilless fabrication of the mH/G UOP system. All the mH/G UOP materials with 1 Mol% (in mole percent) guest dispersed in hosts exhibited intense yellow phosphorescence after the stoppage of 365 nm UV irradiation (Fig. 1d). Powder X-ray diffraction (PXRD) studies for the host matrices and mH/G UOP system were carried out, as shown in Supplementary Fig. 5. The PXRD data for the mH/G UOP materials were similar to the relative host matrices. These studies proved that the crystalline structures of the host matrices were unchanged after doping 1 Mol% PzPh. Thus, the observation of afterglows in the mH/G UOP system could be attributed to the addition of guest molecules other than morphology changing. In addition, thermal analyses for the guest, hosts and mH/G UOP materials were carried out (Supplementary Figs. 6-8 and Supplementary Table 3). According to the TGA (thermogravimetry analyses) studies for the guest and various hosts, decent thermal stabilities of the mH/G UOP materials could be ensured. The DSC (differential scanning calorimeter) analyses showed that different melting points of the mH/G UOP materials could be achieved by switching the host materials.

**Photophysical properties of the mH/G UOP system.** Photophysical investigations of crystalline PzPh and mH/G UOP system

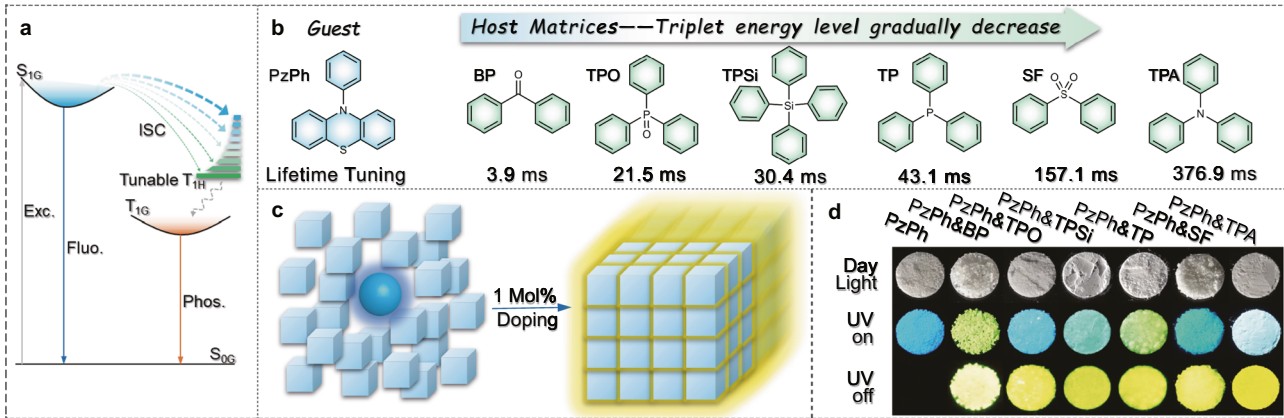

**Fig. 1 Design and fabrication of mH/G UOP system. a** Jablonski diagram for ISC and energy transfer pathways between host and guest species of the lifetime-tunable mH/G UOP system (Exc., Fluo., and Phos. are referred to Excitation, Fluorescence, and Phosphorescence). **b** chemical structures of the guest emitter PzPh and host matrices (BP, TPO, TPSi, TP, SF, and TPA) and lifetimes of the mH/G UOP system. **c** Schematic illustration of mH/G doping strategy. **d** Photographs of PzPh and mH/G UOP materials at the conditions of "Daylight", "UV on (365 nm)" and "UV off".

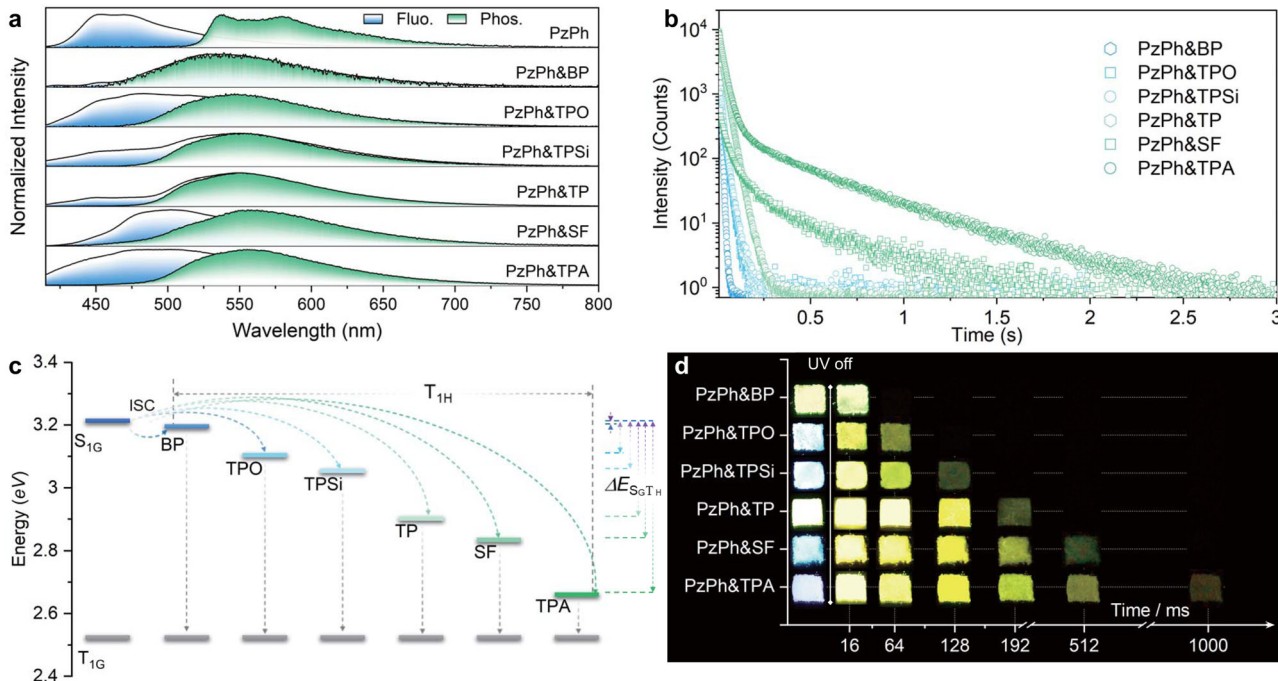

**Fig. 2 Photophysical properties of crystalline powders for PzPh and the mH/G UOP system under 365 nm irradiation at room temperature. a** Steady-state PL (blue filled, $\lambda_{ex} = 365$ nm; PzPh&TPA was measured under 400 nm irradiation) and delayed spectra (green filled, delayed time = 8 ms; PzPh was measured at 77 K). **b** Decay curves at the phosphorescence emission band of 543–556 nm. **c** The energy levels of the hosts and PzPh. **d** Photographs taken under a 365 nm UV lamp on and off.

were performed. The corresponding data have been summarized in Fig. 2 and Supplementary Table 1. Phosphorescence of PzPh (obtained at 77 K) was detected as a structured band ($\upsilon = 1395$ cm$^{-1}$) with the emission maximum of 534 nm. After the stoppage of UV irradiation, bright-yellow afterglow was observed in the mH/G UOP system. By switching host matrices, the afterglow emission colors for the mH/G UOP system were almost unchanged with the CIE coordinates in the range of (0.39 – 0.42, 0.52 – 0.55) (Supplementary Fig. 9). As shown in Fig. 2a, delayed spectra of all the mH/G UOP materials featured the phosphorescence emission bands at ca. 550 nm, which were similar in energy to that of phosphorescence band of PzPh, indicating that phosphorescence of the doping system was generated from the triplet excitons of the guest emitter. Besides, the emission maxima of the UOP were almost maintained with various

host matrices, which further confirmed the origin of UOP in the doping system. Steady-state PL spectra of these mH/G UOP materials consisted of fluorescence and phosphorescence of PzPh, resulting in a linear arrangement of the corresponding CIE chromaticity coordinates between the two luminescent components of PzPh. As a consequence, such a facile mH/G doping strategy can also be used to achieve color-tunable luminescence in the field of optoelectronics.

To further investigate the photophysical properties of the mH/G UOP system, time-resolved decay curves of these mH/G UOP materials were performed and shown in Fig. 2b. All these mH/G UOP materials exhibited two-exponential lifetimes under the 365 nm UV excitation (PzPh&TPA was measured under 400 nm excitation exceptionally to avoid the fluorescence of TPA from being excited, Supplementary Fig. 18). Their lifetimes consisted of

a shorter component of ca. 10–25 ms and another continuously tunable component. The tunable component could be gradually enhanced from 3.9 to 376.9 ms by enlarging the $\Delta E_{SGTH}$. The host matrices could provide a rigid environment to restrain the motion of guest molecule. In addition, water and oxygen, which may quench the triplet emission, could also be blocked. Therefore, they all had a decay component with similar lifetimes, corresponding to the shorter lifetime of about tens of milliseconds aforementioned. The other decay component with adjustable lifetimes were attributed to inconsistent energy transfer caused by continuously increasing $\Delta E_{SGTH}$ between host and guest species, as shown in Fig. 2c. By comparing the phosphorescence lifetimes of the mH/G UOP system and their relative host matrices, no obvious relationship could be summarized (Supplementary Table 1 and 2). Thus, the wide-range lifetime-tunable properties were not achieved by changing the phosphorescence lifetimes of the host matrices. Different afterglows between the mH/G UOP system and the host materials (Supplementary Fig. 19) further demonstrated the different phosphorescent originations for the mH/G UOP system and the host materials. On the contrary, the phosphorescence lifetimes of the mH/G UOP system were perfectly in line with $\Delta E_{SGTH}$ as shown in Supplementary Table 1 and 2. Therefore, it should be noted that the turnability of the mH/G UOP system was achieved by manipulating the $T_{1H}$ of the host matrices and controlling the ISC processes. For instance, the fluorescence band of PzPh showed a large overlap with the phosphorescence band of BP, providing a very small $\Delta E_{SGTH}$ for PzPh&BP. As a consequence, a rapid and efficient ISC occurred between the host and guest species, resulting in a fast (lifetime of 3.9 ms) and dominating phosphorescence decay component (99.2%). On the contrary, the $\Delta E_{SGTH}$ of PzPh&TPA, (ca. 0.54 eV) was much larger than that of PzPh&BP. The large $\Delta E_{SGTH}$ of PzPh&TPA resisted the ISC process from singlet state (guest) to triplet state (host). Thus, a relative slow and nondominant UOP-decay component (34.2%) with longer lifetime of 376.9 ms was generated in PzPh&TPA. By tuning the triplet energy of the host material, the lifetime could be enhanced almost 100 folds. Regularly, the remaining four mH/G UOP materials showed an increasing lifetime from 21.5 to 157.1 ms along with the gradually increased $\Delta E_{SGTH}$. By taking advantage of the wide-range tunable lifetime properties of the mH/G UOP system, duration of the bright-yellow afterglows could be manipulated from about 0.03 to 2.2 s in ambient conditions. Figure 2d showed the luminescence (under 365 nm UV irradiation) and afterglow (stop UV irradiation) photographs of the mH/G UOP system. It showed that there were well-defined boundary nodes for the afterglow quenching time in each host/guest material, reflecting the superior performance of these wide-range lifetime-tunable UOP materials applied in fields of precise and strict anticounterfeiting and information encryption.

**Excitation-dependent UOP properties of the mH/G UOP system.** In particular, these mH/G UOP materials show excitation-dependent UOP at room temperature (Fig. 3, Supplementary Figs. 14-18). As shown in Fig. 3a, great enhancement arose in the phosphorescence lifetimes of PzPh&TPO, PzPh&TPSi, and PzPh&SF under the excitation of 280 nm UV light, compared to those excited by lower-energy UV excitation (365 nm). PzPh&TPA also showed an inconspicuously increased lifetime when the excitation light was blue-shifted from 400 to 365 nm (Supplementary Fig. 18). It was assumed that the long-lived triplet excitons of host matrices played important roles in prolonging the UOP lifetimes of the mH/G UOP system. Excitation-dependent decay curves of PzPh&TPSi were measured and shown in Fig. 3b. Compared with the UOP excited at 365 nm,

an additional long-decay component with a lifetime up to 778.9 ms was observed under 280 nm excitation. This long-lived component might be attributed to the exciton transitions from the singlet state to the lowest triplet state of TPSi via ISC and internal conversion, which then transfer to the triplet state of PzPh through energy transfer to participate in the UOP processes. In the mH/G UOP system, triplet-exciton involved energy transfers were proposed as the short-range Dexter mechanism (<1 nm) according to relative literatures[40–43]. To illustrate the energy transfer process in PzPh&TPSi, the absorption, phosphorescence excitation and emission spectra of the PzPh and TPSi were performed in Supplementary Figs. 21 and 22. PzPh exhibited a broad absorption band with the maximum of 370 nm in crystalline state. TPSi showed a narrow absorption band with the maximum of 269 nm, indicating a larger energy gap. In addition, the phosphorescence excitation spectrum of PzPh (at phosphorescence peak 534 nm) was overlapped with the phosphorescence emission spectrum of TPSi in a certain extent. These results demonstrated the possibility of Dexter energy transfer between the host and guest species. Dexter energy transfer was sensitive to the concentration of guest material according to previous literatures[42,43]. As shown in Supplementary Figs. 23 and 24, the relative phosphorescence intensity of PzPh&TPSi (at 549 nm) showed an obvious enhancement and the time-resolved decay curves were shortened as the concentration of PzPh increased. These results indicated a higher energy transfer efficiency with the increasing PzPh doping ratio, and further demonstrated the Dexter energy mechanism for the mH/G UOP system. To investigate the excitation origin of the phosphorescence in PzPh&TPSi, the excitation spectra of PzPh, TPSi, and their doping system are studied. As shown in Fig. 3c, the excitation spectrum of phosphorescence for PzPh&TPSi exhibited a similar shape to that of TPSi before 305 nm excitation (blue filled region), meanwhile, the remaining part showed a similar trend as PzPh after 305 nm excitation (green filled region). Comparable patterns can also be found in the other five mH/G UOP materials (Supplementary Figs. 25–29). It revealed that the phosphorescence in the doping system resulted from the absorption of excitation energy by host matrices under high-energy excitation, and was determined by the excited-state energy transfer of the guest emitter in the case of low-energy excitation. Phosphorescence of the mH/G UOP system could be generated in multiple pathways by energy transfer between host and guest species. Excitation-emission mapping of PzPh&TPSi revealed the synergistic relationship between the phosphorescence of host/guest system and fluorescence of the host and guest species clearly (Fig. 3d). The formation of UOP was always accompanied by the generation of host or guest fluorescence. Therefore, UOP systems with long lifetimes and high efficiencies were expected to be obtained under higher-energy UV irradiation due to multi-ISC pathways to form long-lived triplet excitons. The UOP spectra of the hosts and the mH/G UOP materials under different excitation wavelengths (280 nm, 365 nm) were shown in Supplementary Figs. 30–35. For other host materials except for TP and TPA, obvious phosphorescence bands could be detected under higher-energy excitation. In the phosphorescence spectra, mH/G UOP materials and their relative host materials showed obvious differences in emission maxima, spectral structures and decay lifetimes (Fig. 3a, Supplementary Fig. 12). In addition, under high-energy excitation (280 nm), some triplet excitons of the host matrices contributed to phosphorescence through radiation transition. Thus, the UOP spectra consisted of the phosphorescence from the mH/G system (main part) and the host species (a small part) were slightly broadened. Afterglow color changes might be observed in PzPh&TPSi and PzPh&SF (Supplementary Figs. 32 and 34). Excitation-dependent UOP properties (showing great enhanced

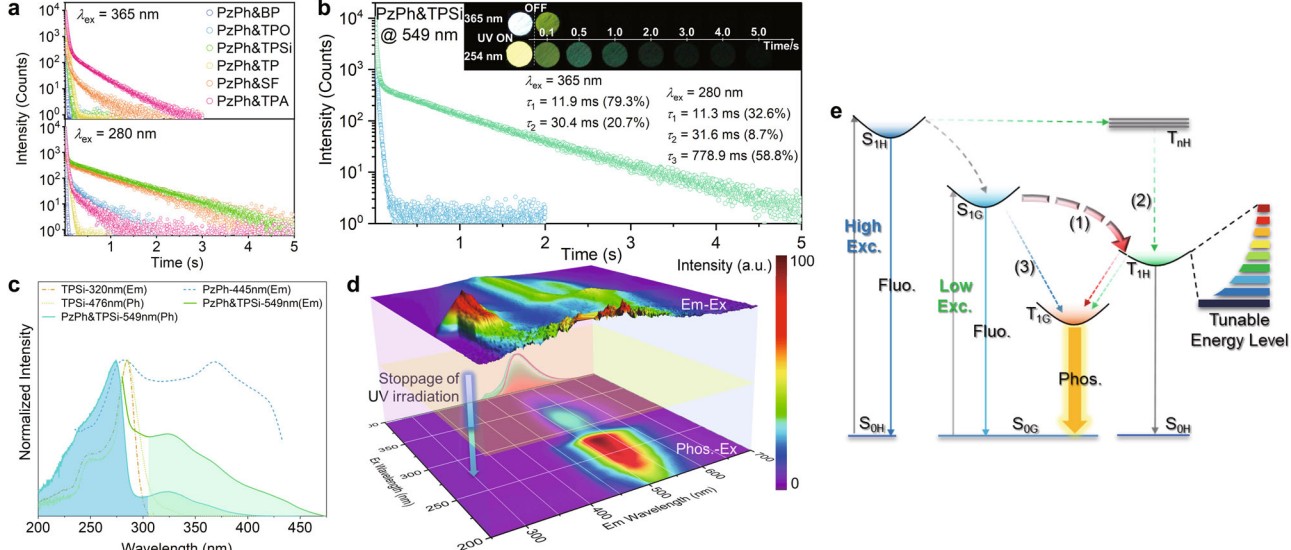

**Fig. 3 Excitation-dependent UOP properties and UOP mechanism of the mH/G UOP system. a** Time-resolved decay curves for the mH/G UOP system under 365 nm and 280 nm excitation. **b** Time-resolved decay curves of PzPh&TPSi under 365 nm and 280 nm excitation (inset: Photographs taken under a 365 or 254 nm UV lamp on and off). **c** Excitation spectra of PzPh, TPSi, and their doping system ("Em" was referred to the steady-state emission peak, and "Ph" was referred to the phosphorescence emission peak). **d** Excitation-emission mapping of PzPh&TPSi under ambient conditions (top: excitation-steady-state emission mapping; bottom: excitation-delayed emission mapping, the Rayleigh scattering was removed by Delaunay triangulation method[44–46]; side: phosphorescence spectra taken under 365 nm (red line) and 280 nm (green line) excitation. **e** Jablonski diagram for proposed photophysical processes between host and guest species in the mH/G UOP system.

lifetimes for some mH/G materials under high-energy excitation) could further enrich the variability of the mH/G UOP system.

**UOP mechanism study.** To clarify the phosphorescence lifetime-tuning strategy and gain a deeper understanding of the UOP mechanism in the mH/G UOP system, the Jablonski diagram for proposed photophysical processes between host and guest species were summarized in Fig. 3e. There are mainly three pathways contributing to the multicomponent UOP. First of all, the lowest triplet state of host matrices ($T_{1H}$) took great advantages to form long-lived phosphorescence by affording an energy-tunable intermediate platform to boost ISC and energy transfer processes (path (1), red dotted arrows) under low-energy excitation. Based on the energy-tunable intermediate platform, phosphorescence lifetimes could be easily manipulated in a wide range by adjusting the difficulty and efficiency of ISC and energy transfer via various $\Delta E_{SGTH}$. In addition, the host matrices could be excited under high-energy irradiation and constructed an additional path to form UOP with a prolonged lifetime through intramolecular ISC process (path (2), green dotted arrows), which endowed the mH/G UOP system excitation-dependent UOP lifetime properties in ambient conditions. As a complement, the rigid environment of host matrices could restrain the non-radiative energy dissipation of guest excited states effectively and cause a competitive pathway for the formation of phosphorescence via the intramolecular ISC process of the guest emitter (path (3), blue dotted arrow).

**Applications.** Fig. 4 schematically illustrates the applications of the lifetime-tunable mH/G UOP system in anticounterfeiting, information displaying, optical storage, and security protection. All the thin films were fabricated by a screen-printing method, in which doping materials were printed on light-absorbing flexible aluminum foil substrates. Patterns on the substrates were combined with these mH/G UOP materials in different host matrices

as shown in Fig. 4a. Thus, patterns spliced with the same afterglow color but tremendous variation in lifetimes were fulfilled. Benefiting from the melting points feature of host matrices (Supplementary Table 3), selective afterglow quenching or promoting in the mH/G UOP system could be controlled by temperature. As shown in Fig. 4b, the flexible thin film exhibited intense yellow afterglow with the Chinese characters "遇見" after a stoppage of 365 nm UV lamp irradiation at room temperature. As the temperature increased, host matrices with low melting points gradually melted and the corresponding afterglows disappeared, resulting in different Chinese characters at different temperatures. When it returned back to room temperature, the lost afterglows reappeared along with the initial Chinese characters, providing potential application in thermosensitive anticounterfeiting. Fig. 4c showed the excitation-dependent UOP characteristic with the mH/G UOP system, the pattern displayed different dynamic information (UOP lifetime and color) after the stoppage of UV light irradiation with different wavelengths. These dynamic-response characteristics of afterglow varying with the change of ambient conditions can be exploited for dynamic multidimensional anticounterfeiting. By taking advantages of the characteristics of distinguishable time-resolved triplet excitons decay, we can also store large amounts of optical information in a time carrier, such as afterglow with "B8", "A9", "P3", "F3", "C7", and "L1" exhibited within 1 s in a simply fabricated thin film. It is expected to develop into a new generation of time-resolved information displaying, realizing an ultrasmall size and superlarge capacity. Furthermore, we can hide the encrypted information at a certain time node to achieve more sophisticated and higher-level information encryption and security protection. As demonstrated in Fig. 4e, real information was hidden in a series of fake messages according to the wide-range lifetime-tunable property of the mH/G UOP system. The encrypted information "IFE" could only be identified at about 200 ms (before reading the information, desaturated and reversed-phase processes should be carried out).

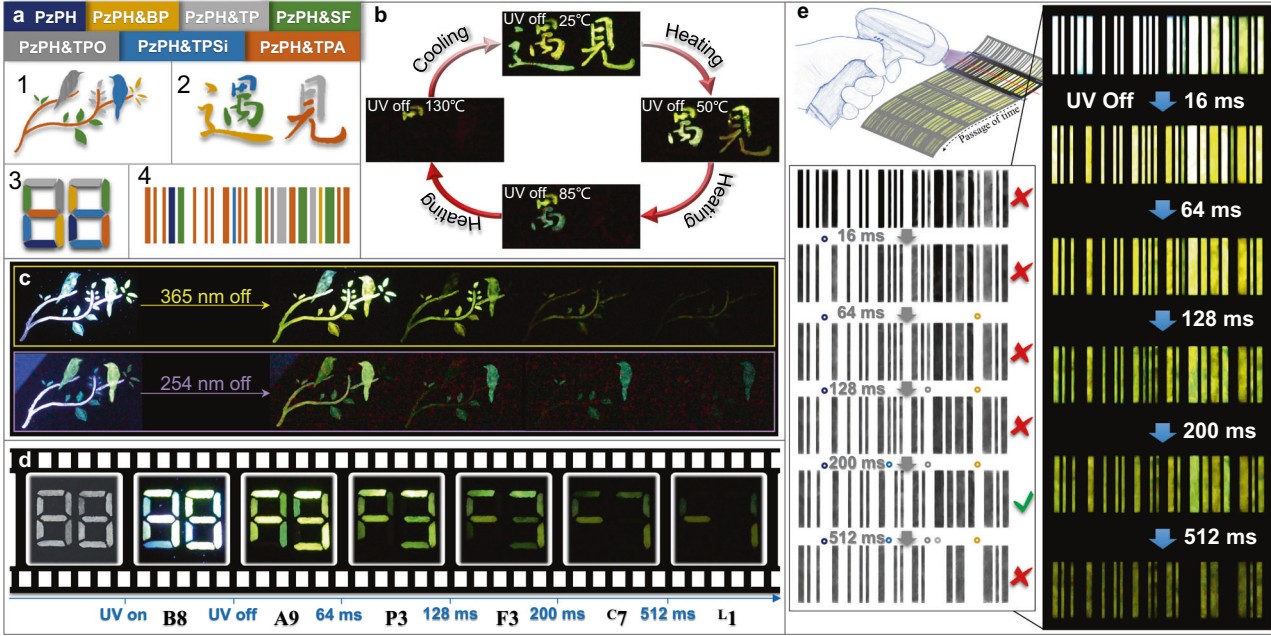

**Fig. 4 Schematic illustrations of lifetime-order encoding for applications using the mH/G UOP system. a** screen-printing patterns adopted for flexible thin films. **b** The UOP photographs taken after the stoppage of 365 nm excitation for thermosensitive anticounterfeiting by varying temperatures from 25 to 130 °C. **c, d** The UOP images along with duration time after the stoppage of UV light irradiation with different wavelengths (254 nm or 365 nm for **c**, and 365 nm for **d**). **e** Dynamic afterglow bars with different duration time for higher-level information encryption.

## Discussion

In conclusion, we have proposed a concise and guiding mH/G doping strategy to achieve UOPs with wide-range tunable lifetimes based on PzPh and various host matrices. The rigidity, proper triplet energy level, and efficient ISC process of the host molecules increase the triplet harvesting in three possible pathways to provide bright-yellow UOP with decent phosphorescence efficiencies. In addition, the phosphorescence lifetimes could be rationally tuned over a wide range from 3.9 to 376.9 ms with stable afterglow color in ambient conditions. This facile approach can also provide a controllable afterglow responding to various UV irradiation and temperatures. Based on this wide-range lifetime-tunable mH/G UOP system, more sophisticated and higher-level information encryption and dynamic multidimensional anticounterfeiting can be achieved easily based on the lifetime-order encoding characteristic and stimulus-response diversity. In general, combined advantages of stability, low cost, simplicity, and excellent performances (excitation dependence, temperature dependence, and time-resolved emission properties), these mH/G UOP materials are expected to attract widespread attention and favors from the fields of material science and flexible electronics.

## Methods

**Preparation of host/guest doping systems and flexible thin films**. Materials: The guest emitter PzPh was synthesised according to ref. [37], and the host molecules were purchased from damas-beta (TPO, TP, TPA), aladdin (BP, SF), and TCI (TPSi), respectively. All the chemicals were purified by column chromatography then followed by recrystallization. Preparation of the mH/G UOP materials with Solution evaporation method: 1.00 g of host matrix (BP, TPO, TPSi, TP, SF, or TPA) was dissolved in dichloromethane and mixed with the PzPh (1 Mol%). The mixed solution was concentrated by slow rotary evaporation at room temperature, then a series of white crystalline powders were obtained. Melt-casting method: 1.00 g of host matrix and 1 Mol% PzPh were mixed in a degassed round flask, and pumped nitrogen three times before heated up. Slowly cooled down to room temperature after the samples melted completely and achieved the white solid solution of the mH/G UOP materials (except for PzPh&TPSi). Grinding method: to an agate mortar, 1.00 g of host matrix and 1 Mol% PzPh were mixed with 0.5 mL dichloromethane. Continued to grind gently until the white crystalline powders were precipitated to give the mH/G UOP materials. Preparation of flexible thin films: 5 wt% doping materials were uniformly dispersed in aloe vera gel to make transparent colloidal inks. The colloidal inks were coated on the screen-printing template with different patterns. Flexible thin films were achieved by gently scraping and printing information on a light-absorbing aluminium foil substrate.

**Photophysical measurements**. The steady-state photoluminescence (PL) spectra, excitation-emission spectra, delayed emission spectra, and phosphorescence lifetime were measured on a Hitachi F-7100 fluorescence spectrophotometer and an Ocean Optics Spectrometer (QE 65Pro). Fluorescence lifetime and delayed emission spectra at different wavelengths were measured using an Edinburgh FLSP 980 fluorescence spectrophotometer equipped with a laser light source (370 nm). Absolute PL quantum yields were measured on a Hamamatsu absolute PL quantum yield spectrometer C11347-11 Quantaurus-QY (The fluorescence and phosphorescence quantum yields of the multi-host/guest UOP system were calculated from the proportion in their steady-state emission spectra and phosphorescence spectra). Photos and videos were recorded by a Sony LICE-6400M camera.

## Data availability

The X-ray crystallographic coordinates for structures reported in this study are downloaded from the Cambridge Crystallographic Data Centre (CCDC), under deposition numbers 1135756 (PzPh), 245188 (BP), 1275222 (TPO), 1275211 (TPSi), 130547 (TP), 745918 (SF), 660790 (TPA). These data can be obtained free of charge from The Cambridge Crystallographic Data Centre via www.ccdc.cam.ac.uk/data_request/cif. The data that support the plots within this paper and other findings of this study are available from the corresponding author upon reasonable request.

## Code availability

No custom computer code is used in the manuscript.

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

## Acknowledgements

We gratefully acknowledge the financial support from the NSF of China (51703253), the Fundamental Research Funds for the Central Universities, Key Research and Development Program of Shaanxi Province (2020GXLH-Z-010), Shaanxi science and technology Fund (2020JQ-168), Pearl River Nova Program of Guangzhou (201906010091), Chongqing science and technology Fund (cstc2020jcyj-msxmX0931), China Postdoctoral Science Foundation (2020M673479), and support from China Aerospace Science and Industry Corporation (2020-HT-XG). Thanks to C. L. Tan for affording the Delaunay triangulation method to eliminate the Rayleigh scattering in 3D excitation-emission mapping with Origin software.

## Author contributions

Z.X., T.Y., and W.H. conceived the idea and designed experiments. Z.X., X.Z., L.J., Z.A., and W.H. wrote the manuscript. Z.X., X.Z., H.W., C.H., H.S., and M.D. were primarily responsible for the experiments. X.Z. and H.S. conducted the lifetime measurements. H.W. and M.D. measured the quantum efficiency and performed the spectra measurements. C.H. took the photographs and videos. All authors contributed to the data analyses.

## Competing interests

The authors declare no competing interests.
