## [Peer Review File · Nature Communications]

Reviewer #1 (Remarks to the Author):

Manuscript "Wide-range Lifetime-tunable and Responsive Ultralong Organic Phosphorescent multi-Host/Guest System" reports on a series of host/guest (10-phenylphenothiazine) materials displaying multi-stimuli (temperature and excitation wavelength) UOP features. The prepared systems are fully optically characterized and very nicely exploited in anti-counterfeiting, information displaying, optical storage, and security protection. I found the subject potentially very interesting and inspiring for a broad number of readers.

However, even though the work is fully consistent from the photophysical point of view, it needs implementation regarding the materials' thermal and PXRD characterization. Moreover, the English used must be reassessed through the text.

I suggest the following improvements/modifications:

- 1) In the Abstract: add RT to "non-phosphorescent emitter"
- 2) Introduction line 62: sentence "However, the main role of the host matrices playing in the host/guest system remains unclear, it is often unsuitable for all rigid matrices to a guest emitter neither. " is not clear and should be reformulated.
- 3) Results: sentences "PzPh was chosen as the guest emitter for its suitable ΔE_{ST} (ca. 0.7 eV) and good compatibility. Pure PzPh has only a blue fluorescent band at 445 nm under UV excitation, no phosphorescence was observed at room temperature. It adopts a quasi-equatorial conformation with a butterfly-shape phenothiazine moiety folded in a dihedral angle of $150.28^{\circ} \sim 162.44^{\circ}$, in which the phenyl ring is orthogonally connected to the phenothiazine (Supplementary Fig. 2)." Need proper references
- 4) I would use a single brief statement to refer to the materials through the text. In the present versions they are indicated as: "These doping materials", "multi-host/guest doping system", "host/guest UOP materials", "multi-Host/Guest System" "multi-host/guest UOP system"
- 5) Color codes in Figure 3b are unreadable
- 6) English must be improved: "Photophysical properties of crystalline PzPh and host/guest UOP materials were studied, the corresponding data were shown in Fig. 2 and Supplementary Table 1." "Their lifetimes contain a shorter component of ca. 10 ~ 25 ms and another continuously tunable component which gradually increasing from 3.9 ms to 376.9 ms, respectively." "Based on the wide-range tunable lifetimes of the multi-host/guest system, duration of the bright-yellow afterglows could manipulate from about 0.03 s to 2.2 s in ambient conditions." "This led to the broadening of phosphorescence band in the doping systems, which was finally caused change of afterglow color" just to mention few examples
- 7) Sentences "Photophysical properties of crystalline PzPh and host/guest UOP materials were studied, the corresponding data were shown in Fig. 2 and Supplementary Table 1. Phosphorescence of the guest emitter PzPh was obtained at 77K with a structured emission bands at 534 nm ($\nu = 1395 \text{ cm}^{-1}$). After the stoppage of UV irradiation, bright-yellow afterglow of host/guest materials with little changes of CIE chromaticity coordinates within (0.39 ~ 0.42, 0.52 ~ 0.55) was produced (Supplementary Fig. 4)." Should be reformulated. In addition: are the CIE coordinates reported in Figure S4 referred to PL spectra? Caption to Figure 4 is not clear.
- 8) Sentence "All these host/guest UOP materials 130 exhibit two-exponential lifetimes under the 365 nm UV excitation (PzPh&TPA was measured under 390 nm 131 excitation individually, Supplementary Fig. 13, avoiding the fluorescence of TPA from being excited)." Disagrees with what reported in Caption to Figure 13.
- 9) Legend in Figure 3 c is illegible.
- 10) The experimental section has definitely to be implemented: thermal analysis of the materials

should be added.

11) "Grinding method: to an agate mortar, 1.00 g of host matrix and 1% Mol PzPh were mixed with minority of dichloromethane. Continued to grind gently until the white crystalline powders were precipitated to give the host/guest doping systems." What is the quantitative correspondence of "minority"? if the white crystalline powders precipitate I imagine that at the beginning they are in solution. My opinion is also that XRPD data should be added.

12) "Absolute PL quantum yields were measured on a Hamamatsu absolute PL quantum yield spectrometer C11347-11 Quantaaurus-QY. Photos and videos were recorded by a Sony LICE-6400M camera.": how are fluorescence and phosphorescence QY calculated separately?

Reviewer #2 (Remarks to the Author):

In this work, a series of host/guest UOP materials with dynamic lifetime-tuning properties were reported. And, these materials exhibited excitation-dependent UOP and temperature-controlled afterglow on/off switching properties. The experiments showed that the doping systems have good (UOP) performance and demonstrated application potential in anti-counterfeiting and information encryption. The manuscript is gorgeously illustrated, but the conclusions are overstated because the photophysical properties of the host molecules are incomplete, and it is not fully compared with the host/guest system. In addition, there are many subtle errors in the manuscript. The paper might be reconsidered for possible publication after a major revision.

1. In Figure S13, the excitation wavelength is 400 nm, but in the text the author says the $\lambda_{ex} = 390$ nm. So, which one is correct?

2. Line 169, the author describes the (lifetime) excitation dependence of PzPh&TPA from 390 to 365 nm, but there is no PzPh&TPA spectrum in Figure S9. Is the author describing the PzPh&BP?

3. Again, in Table S1, the excitation wavelength of PzPh&TPA has changed to 365/390 nm, which is inconsistent with the 400 nm stated in the text. Please revise it! Similar errors appear repeatedly. If these data are all in a muddle, the author had better be able to retest to ensure the reliability of the results.

4. In Figure 3b, when the $\lambda_{ex} = 280$ nm, the author obtains two lifetimes marked as " τ_2 ". Judging from the Table S1, it should be " $\tau_3 = 778.9$ ms".

5. Line 176-179, the author described that the host TPSi and PzPh&TPSi have similar phosphorescence excitation spectra. Can the energy transfer of the doping system be fully proved by the excitation spectra alone? As far as I know, the possibility of energy transfer needs to be evaluated by the degree of overlap of the absorption spectra of the host and guest materials (because it involves triplet excitons, Figure 3e). In any case, further experiments are necessary to ensure that the energy transfer process proposed in the article is reliable.

6. It is recommended to add an annotation in Figure 3d: PzPh&TPSi. At the same time, the Rayleigh scattering is so strong that it overwhelms the fluorescence. Removing the Rayleigh scattering signal may be an effective option.

7. "Wide-range" and "Lifetime-tunable" appear in the title of the manuscript. Reading through the full text, we find:

1) This tunability is achieved by replacing the host molecule, and these host materials correspond to different phosphorescence lifetimes. Different UOP lifetimes are intrinsic properties of the host materials, not unique to the host-guest doping systems. Therefore, is it really appropriate to put "Lifetime-tunable" in the title? It's worth revisiting!

2) The author should provide afterglow photos or videos of the host materials to facilitate readers to compare the luminescence properties before and after doping. From the data presented so far, the biggest achievement of the doping systems is only to ensure the "uniformity" of the UOP color.

3) We note that the author had pointed out the possibility of high-energy excitation, however, we cannot evaluate its correctness before seeing the phosphorescence spectra of the hosts. So, the UOP spectra of the host molecules under different excitation should be supplemented. By comparing these spectra, the author needs to explain the origin of the excitation dependence of these materials. As we all know, if the guest and the host form an excimer, or there is a significant charge transfer state, it may lead to multi-color luminescence. In addition, if excitation dependence is the nature of the hosts, then the author should not emphasize this property again in the manuscript.

Corrections and changes made in response to the referees' comments

Manuscript submitted to *Nature Communications*

Title: Wide-range Lifetime-tunable and Responsive Ultralong Organic Phosphorescent multi-Host/Guest System

Manuscript ID: NCOMMS-20-43059-T by Tao YU, Wei HUANG *et. al.*

We sincerely thank the reviewers for their comments about our submitted article. Corrections and necessary changes have been made according to the reviewers' comments, and are explained as follows:

(Referees' comments: in black; Corrections made by the authors in response to the comments: in blue)

To Referee 1:

Question 1

In the Abstract: add RT to “non-phosphorescent emitter”.

Response to the question 1 of referee 1

Thanks for the good suggestion to avoid ambiguity. We have changed the “non-phosphorescent emitter” with “non-RTP (room temperature phosphorescence) emitter”. This suggestion could make the paper more precise.

Question 2

Introduction line 62: sentence “However, the main role of the host matrices playing in the host/guest system remains unclear, it is often unsuitable for all rigid matrices to a guest emitter neither.” is not clear and should be reformulated.

Response to the question 2 of referee 1

According to the helpful suggestion, we have changed the sentence “However, the main role of the host matrices playing in the host/guest system remains unclear, it is often unsuitable for all rigid matrices to a guest emitter neither.” with “However, the effect of host matrices in doping systems remains unclear. Suitable host species are difficult to be rationally chosen for a certain guest emitter.” (Page 3, Line 62-64).

Question 3

Results: sentences “PzPh was chosen as the guest emitter for its suitable ΔEST (ca. 0.7 eV) and good compatibility. Pure PzPh has only a blue fluorescent band at 445 nm under UV excitation, no phosphorescence was observed at room temperature. It adopts a quasi-equatorial conformation with a butterfly-shape phenothiazine moiety folded in a dihedral angle of 150.28°~ 162.44°, in which the phenyl ring is orthogonally connected to the phenothiazine (Supplementary Fig. 2).” Need proper references.

Response to the question 3 of referee 1

Thanks for the helpful suggestion. To illustrate both the conformational structure and photophysical properties of PzPh, we have added proper references [37-39] to the sentences “It adopts a quasi-equatorial conformation with a butterfly-shape phenothiazine moiety folded in a dihedral angle of 150.28°~ 162.44°, in which the phenyl ring is orthogonally connected to the phenothiazine (Supplementary Fig. 2)”. We would like to thank the reviewer for polishing our paper.

37. Chen, D.-G. *et al.* Phenothiazine scope: Steric strain induced planarization and excimer formation. *Angew. Chem. Int. Ed.* **58**, 13297-13301, (2019).

38. Tanaka, H., Shizu, K., Nakanotani, H. & Adachi, C. Dual intramolecular charge-transfer fluorescence derived from a phenothiazine-triphenyltriazine derivative. *J. Phys. Chem. C* **118**, 15985-15994, (2014).

39. Borowicz, P. *et al.* Nature of the lowest triplet states of 4'-substituted N-phenylphenothiazine derivatives. *Phys. Chem. Chem. Phys.* **2**, 4275-4280, (2000).

Question 4

I would use a single brief statement to refer to the materials through the text. In the present versions they are indicated as: “These doping materials”, “multi-host/guest doping system”, “host/guest UOP materials”, “multi-Host/Guest System” “multi-host/guest UOP system”

Response to question 4 of referee 1

This is a good suggestion to avoid ambiguity. According to the reviewer’s suggestion, we have used uniform brief statements as mH/G UOP system (multi-host/guest UOP system) or mH/G UOP material

(multi-host/guest UOP material) to refer to the materials through the text.

Question 5

Color codes in Figure 3b are unreadable.

Response to question 5 of referee 1

Thanks a lot for the useful suggestion. The color codes become unreadable is mainly because the UOP emission gradually becomes weaker after the stoppage of excitation. To illustrate the phenomena better, we take the picture again with a higher ISO setting. The color codes in Fig. 3b have been replaced with the new codes as Fig. R1 (Fig. 3b) below. Besides, a magnified illustration for Fig. 3b has been listed as Fig. R2 (Supplementary Fig. 49) in Supplementary Information.

Figure R1 (Fig. 3b). Time-resolved decay curves of PzPh&TPSi under 365 nm and 280 nm excitation (inset: Photographs taken under a 365 or 254 nm UV lamp on and off).

Figure R2 (Supplementary Figure 49). Photographs of PzPh&TPSi taken under a 365 or 254 nm UV lamp on and off.

Question 6

English must be improved: “Photophysical properties of crystalline PzPh and host/guest UOP materials were studied, the corresponding data were shown in Fig. 2 and Supplementary Table 1.” “Their lifetimes contain a shorter component of ca. 10 ~ 25 ms and another continuously tunable component which gradually increasing from 3.9 ms to 376.9 ms, respectively.” “Based on the wide-range tunable lifetimes of the multi-host/guest system, duration of the bright-yellow afterglows could manipulate from about 0.03 s to 2.2 s in ambient conditions.” “This led to the broadening of phosphorescence band in the doping systems, which was finally caused change of afterglow color” just to mention few examples

Response to question 6 of referee 1

Thanks for the careful reviewing and useful suggestion to polish our article. We have corrected grammar mistakes and improved the English in our paper. The sentences to be improved and their correction information are listed as Table R1 below.

Table R1 Corrected information of the sentences to be improved in our paper

	Sentences to be improved	Corrected sentences
Line 121-122	Photophysical properties of crystalline PzPh and host/guest UOP materials were studied, the corresponding data were shown in Fig. 2 and Supplementary Table 1.	Photophysical investigations of crystalline PzPh and mH/G UOP system were performed. The corresponding data have been summarized in Fig. 2 and Supplementary Table 1.
Line 138-140	Their lifetimes contain a shorter component of ca. 10 ~ 25 ms and another continuously tunable component which gradually increasing from 3.9 ms to 376.9 ms, respectively.	Their lifetimes consisted of a shorter component of ca. 10 ~ 25 ms and another continuously tunable component. The tunable component could be gradually enhanced from 3.9 ms to 376.9 ms by enlarging the $\Delta E_{S_0-T_1}$.
Line 161-163	Based on the wide-range tunable lifetimes of the multi-host/guest system, duration of the bright-yellow afterglows could manipulate from about 0.03 s to 2.2 s in ambient conditions.	By taking advantage of the wide-range tunable lifetime properties of the mH/G UOP system, duration of the bright-yellow afterglows could be manipulated from about 0.03 s to 2.2 s in ambient conditions.
Line 223-225	This led to the broadening of phosphorescence band in the doping systems, which was finally caused change of afterglow color	Thus, the UOP spectra consisted of the phosphorescence from the mH/G system (main part) and the host species (a small part) were slightly broadened. Afterglow color changes might be observed in PzPh&TPSi and PzPh&SF

		(Supplementary Fig. 32 and 34).
Line 123-126	After the stoppage of UV irradiation, bright-yellow afterglow of host/guest materials with little changes of CIE chromaticity coordinates within (0.39 ~ 0.42, 0.52 ~ 0.55) was produced (Supplementary Fig. 4).	After the stoppage of UV irradiation, bright-yellow afterglow was observed in the mH/G UOP system. By switching host matrices, the afterglow emission colors for the mH/G UOP system were almost unchanged with the CIE coordinates in the range of (0.39 ~ 0.42, 0.52 ~ 0.55) (Supplementary Fig. 9).
Line 90-91	With such a twist conformation, PzPh was not prone to embed into a flat and compact host matrix.	With such a twisted conformation, PzPh was not easily embedded in a flat and compact host matrix.
Line 129-130	Besides, the emission maxima of the UOP were almost maintained by alternating various host matrices, which further demonstrated that the UOP was originated triplet excitons of PzPh.	Besides, the emission maxima of the UOP were almost maintained with various host matrices, which further confirmed the origin of UOP in the doping system.
Line 157-158	It would become more difficult for excitons to intersystem crossing from singlet state of guest to triplet state between the host and guest species.	The large $\Delta E_{S_0T_1}$ of PzPh&TPA resisted the ISC process from singlet state (guest) to triplet state (host).
Line 273-275	Only the afterglow bar reached around 200 ms, after desaturated and reversed-phase processed, the encrypted information “IFE” can be identified by cellphone scanning.	The encrypted information “IFE” could only be identified at about 200 ms (before reading the information, desaturated and reversed-phase processes should be carried out).

Question 7

Sentences “Photophysical properties of crystalline PzPh and host/guest UOP materials were studied, the corresponding data were shown in Fig. 2 and Supplementary Table 1. Phosphorescence of the guest emitter PzPh was obtained at 77K with a structured emission bands at 534 nm ($\nu = 1395 \text{ cm}^{-1}$). After the stoppage of UV irradiation, bright-yellow afterglow of host/guest materials with little changes of CIE chromaticity coordinates within (0.39 ~ 0.42, 0.52 ~ 0.55) was produced (Supplementary Fig. 4).” Should be reformulated. In addition: are the CIE coordinates reported in Figure S4 referred to PL spectra? Caption to Figure S4 is not clear.

Response to question 7 of referee 1

Thanks to the reviewer for polishing our article. Following the helpful suggestion, the sentences

“Photophysical properties of crystalline PzPh and host/guest UOP materials were studied, the corresponding data were shown in Fig. 2 and Supplementary Table 1. Phosphorescence of the guest emitter PzPh was obtained at 77K with a structured emission bands at 534 nm ($\nu = 1395 \text{ cm}^{-1}$). After the stoppage of UV irradiation, bright-yellow afterglow of host/guest materials with little changes of CIE chromaticity coordinates within (0.39 ~ 0.42, 0.52 ~ 0.55) was produced (Supplementary Fig. 4).” were reformulated as “Photophysical investigations of crystalline PzPh and mH/G UOP system were performed. The corresponding data were summarized in Fig. 2 and Supplementary Table 1. Phosphorescence of PzPh (obtained at 77K) was detected as a structured band ($\nu = 1395 \text{ cm}^{-1}$) with the emission maximum of 534 nm. After the stoppage of UV irradiation, bright-yellow afterglow was observed in the mH/G UOP system. By switching host matrices, the afterglow emission colors for the mH/G UOP system were almost unchanged with the CIE coordinates in the range of (0.39 ~ 0.42, 0.52 ~ 0.55) (Supplementary Fig. 9).” (Page 7-8, Line 121-126).

In addition, the CIE coordinates in Supplementary Fig. 9 were referred to the corresponding steady-state emission spectra. To make the statement clearer, the sentence “values of the CIE coordinates were fitted from their corresponding steady-state emission spectra” was added to the caption of Supplementary Fig. 9.

Question 8

Sentence “All these host/guest UOP materials exhibit two-exponential lifetimes under the 365 nm UV excitation (PzPh&TPA was measured under 390 nm excitation individually, Supplementary Fig. S13, avoiding the fluorescence of TPA from being excited).” Disagrees with what reported in Caption to Figure S13.

Response to question 8 of referee 1

Thanks for correcting our mistake. We have checked the original data and make sure that the time-resolved decay curves of PzPh&TPA were measured under 400 nm excitation. To ensure the accuracy of the data, we have repeated the measurement. These two results were listed below and the error is within permission (Fig. R4). Therefore, the “PzPh&TPA was measured under 390 nm excitation” was changed to “PzPh&TPA was measured under 400 nm excitation”. The steady-state PL spectrum of PzPh&TPA under 400 nm excitation was updated in Fig. R3 (Fig. 2a). To improve the English, the sentence “All these host/guest UOP materials exhibit two-exponential lifetimes under the 365 nm UV excitation (PzPh&TPA was measured under 390 nm excitation individually, Supplementary Fig. S13, avoiding the fluorescence of

TPA from being excited).” has been corrected as “All these host/guest UOP materials exhibited two-exponential lifetimes under the 365 nm UV excitation (PzPh&TPA was measured under 400 nm excitation exceptionally to avoid the fluorescence of TPA from being excited, Supplementary Fig. 18).” (Page 8, Line 136-138).

Figure R3 (Fig. 2a). Steady-state PL (blue filled, $\lambda_{ex} = 365$ nm; PzPh&TPA was measured under 400 nm irradiation) and delayed spectra (green filled, delayed time = 8 ms; PzPh was measured at 77 K).

Figure R4 (Supplementary Figure 18). Time-resolved decay curves of PzPh&TPA under 365 nm and 400 nm excitation at room temperature.

Question 9

Legend in Figure 3c is illegible.

Response to question 9 of referee 1

Thanks for the careful reviewing and the useful suggestion. To avoid misunderstanding, different colors with obvious contrast were used in the legend in Fig. 3c.

Figure R5 (Left: original Fig. 3c; Right: revised Fig. 3c) Excitation spectra of PzPh, TPSi, and their doping system (“Em” was referred to the steady-state emission peak, and “phos.” was referred to the phosphorescence peak).

Question 10

The experimental section has definitely to be implemented: thermal analysis of the materials should be added.

Response to question 10 of referee 1

We appreciate the helpful suggestion from the reviewer. Thermal analyses are fundamental data for these mH/G UOP materials to estimate their stabilities. Besides, afterglows for the mH/G UOP system are selectively quenched or promoted according to the various melting points of host matrices. TGA (thermogravimetry analyses) were performed for the guest and several hosts respectively (Fig. R8 (Supplementary Fig. 8)). In addition, DSC (differential scanning calorimeter) studies were performed for the hosts and the mH/G UOP materials (Fig. R6 and R7 (Supplementary Fig. 6 and 7)). The results of these DSC and TGA were summarized in Table R2 (Supplementary Table 3).

To improve the paper, a paragraph of “In addition, thermal analyses for the guest, hosts and mH/G UOP materials were carried out (Supplementary Fig. 6-8 and Supplementary Table 3). According to the TGA (thermogravimetry analyses) studies for the guest and various hosts, decent thermal stabilities of the mH/G UOP materials could be ensured. The DSC (differential scanning calorimeter) analyses showed that

different melting points of the mH/G UOP materials could be achieved by switching the host materials.”
has been added on page 6 (Line 108-112).

Figure R6 (Supplementary Figure 6). DSC curves of PzPh and host materials (BP, TPO, TPSi, TP, SF and TPA) under N₂ atmosphere at 10 °C/min heating rate.

Figure R7 (Supplementary Figure 7). DSC curves of the mH/G UOP system under N₂ atmosphere at 10 °C/min heating rate.

Figure R8 (Supplementary Figure 8). TGA curves of PzPh and host materials (BP, TPO, TPSi, TP, SF and TPA) under N₂ atmosphere at 20 °C/min heating rate.

Table R2 (Supplementary Table 3). Thermal properties of crystalline powders for host materials and PzPh.

Sample	BP	TPO	TPSi	TP	SF	TPA	PzPh
T _m (°C)	52.3	159.5	241.4	83.1	127.6	128.5	97.1
T _d (°C)	115.7	233.6	246.0	176.9	201.7	175.0	203.6

* melting temperature (T_m) and decomposition temperature (T_d) of 5% weight loss under N₂ atmosphere at 10 °C/min (DSC) and 20 °C/min (TGA) heating rate.

Question 11

“Grinding method: to an agate mortar, 1.00 g of host matrix and 1% Mol PzPh were mixed with minority of dichloromethane. Continued to grind gently until the white crystalline powders were precipitated to give the host/guest doping systems.” What is the quantitative correspondence of “minority”? if the white crystalline powders precipitate, I imagine that at the beginning they are in solution. My opinion is also that XRPD data should be added.

Response to question 11 of referee 1

The reviewer provided us with a good suggestion. The previous statement is not clear enough and misleading. In the preparation of the mH/G UOP system, 1.00 g host material and 1 Mol% PzPh were

mixed together with *ca.* 0.5 mL dichloromethane. The beginning of the mixture was in paste state not in solution state. During the grinding process, the dichloromethane evaporated and the crystalline powders formed. To describe the grinding method more accurately, the sentence “**Grinding method:** to an agate mortar, 1.00 g of host matrix and 1% Mol PzPh were mixed with minority of dichloromethane. Continued to grind gently until the white crystalline powders were precipitated to give the host/guest doping systems.” was replaced with “**Grinding method:** to an agate mortar, 1.00 g of host matrix and 1 Mol% PzPh were mixed together with *ca.* 0.5 mL dichloromethane. The pasty mixture was ground gently to form white crystalline powders as the mH/G UOP materials.” (Page 17, Line 306-308). In addition, the words “1% Mol PzPh” were replaced with “1 Mol% PzPh”.

Following the reviewer’s suggestion, the Powder X-ray diffraction (PXRD) data were measured for these host matrices and mH/G UOP materials. The comparison of PXRD data for host matrices and the dopants mH/G UOP materials were shown as the following and added in Supplementary Fig. 5. These studies proved that the crystalline structures were almost unchanged after doping 1 Mol% PzPh. Without aggregated state changing, the observation of afterglows in the mH/G UOP system could be attributed to the addition of guest molecules.

To improve our paper, a paragraph was added in page 6 (Line 104-108) in the paper as following: “Powder X-ray diffraction (PXRD) studies for the host matrices and mH/G UOP system were carried out, as shown in Supplementary Fig. 5. The PXRD data for the mH/G UOP materials were similar to the relative host matrices. These studies proved that the crystalline structures of the host matrices were unchanged after doping 1 Mol% PzPh. Thus, the observation of afterglows in the mH/G UOP system could be attributed to the addition of guest molecules other than morphology changing.” We would like to give thanks for the careful reviewing and helpful suggestions to improve the logicity and the rigor of the paper.

Figure R9 (Supplementary Figure 5). PXRD patterns of the mH/G UOP materials and their corresponding host and guest species.

Question 12

“Absolute PL quantum yields were measured on a Hamamatsu absolute PL quantum yield spectrometer C11347-11 Quantaaurus-QY. Photos and videos were recorded by a Sony LICE-6400M camera.”: how are fluorescence and phosphorescence QY calculated separately?

Response to question 12 of referee 1

We really appreciate the useful question from the reviewer. The method to calculate the fluorescence and phosphorescence quantum yields separately was performed according to previous literatures [1, 2]. In these literatures, the phosphorescence bands could be obtained in their delayed emission spectrum. According to the structure of phosphorescence bands, the fluorescence and phosphorescence emission bands could be separated in steady state emission spectra. The ratio for fluorescence and phosphorescence quantum yields could be calculated with areas of separated fluorescence and phosphorescence bands. Thus, the fluorescence and phosphorescence quantum yields could be obtained with their total luminescence quantum yields and the ratio for the two relative quantum yields, as shown in Table R3 (updated data for second measurement).

To avoid confusion, a paragraph of “The fluorescence and phosphorescence quantum yields of the mH/G UOP system could be calculated separately from the total luminescence quantum yields according to the

method in previous literatures.^{1, 2} has been added on Supplementary Table 1 to illustrate the relative references which provide the method to calculate the fluorescence and phosphorescence quantum yields separately.

Table R3. Quantum Yields of crystalline powders for the mH/G UOP materials.

Quantum Yields	PzPh&BP	PzPh&TPO	PzPh&TPSi	PzPh&TP	PzPh&SF	PzPh&TPA
Φ_{PL} (%)	10.3	2.4	3.3	10.2	10.1	16.2
Φ_F (%)	0.9	1.1	0.9	1.3	5.7	9.3
Φ_P (%)	9.4	1.3	2.4	8.9	4.4	6.9

Table R4 (Supplementary Table 1). Photophysical properties of crystalline powders for PzPh and the mH/G UOP materials.

Sample	λ_{ex} (nm)	Fluo.			Phos.				
		λ_F (nm)	Φ_F (%)	τ_F (ns)	λ_P (nm)	Φ_P (%)	τ_{P1} (ms)	τ_{P2} (ms)	τ_{P3} (ms)
PzPh	365	445; 472	2.1	3.7	534; 577	-	-	-	-
PzPh&BP	365	445	0.9	3.7(445 nm)	543	9.4	10.6 (0.8%)	3.9 (99.2%)	-
	280	445	-	-	543	-	10.2 (3.4%)	4.8 (96.6%)	-
PzPh&TPO	365	445; 477	1.1	3.8(445 nm)	547	1.3	10.3 (29.3%)	21.5 (70.7%)	-
	280	292; 445	-	-	547	-	10.6 (19.3%)	20.5 (72.4%)	257.0 (8.2%)
PzPh&TPSi	365	445; 472	0.9	3.7(445 nm)	549	2.4	11.9 (79.3%)	30.4 (20.7%)	-
	280	305; 451	-	-	450; 549	-	11.3 (32.6%)	31.6 (8.7%)	778.9 (58.8%)
PzPh&TP	365	445; 475	1.3	3.0(445 nm)	547	8.9	23.3 (89.8%)	43.1 (10.2%)	-
	280	445	-	-	547	-	22.7 (84.4%)	43.2 (15.6%)	-
PzPh&SF	365	472; 498	5.7	5.6(445 nm)	556	4.4	22.6 (56.9%)	157.1 (43.1%)	-
	280	341; 492	-	-	494; 556	-	22.5 (6.7%)	154.4 (5.2%)	836.2 (88.1%)
PzPh&TPA	400	445; 477	9.3	3.0(445 nm)	555	6.9	25.4 (65.8%)	376.9 (34.2%)	-
	365	390	-	-	400; 555	-	25.0 (70.0%)	428.1 (20.8%)	83.3 (9.2%)

* The fluorescence and phosphorescence quantum yields of the mH/G UOP system could be calculated separately from the total luminescence quantum yields according to the method in previous literatures.^{1, 2}

1. Yang, Z. *et al.* Boosting quantum efficiency of ultralong organic phosphorescence up to 52% via intramolecular halogen bonding. *Angew. Chem. Int. Ed.* **58**, 17451-17455, (2020).
2. Lei, Y. *et al.* Wide-range color-tunable ultralong organic phosphorescence materials for printable and writable security inks. *Angew. Chem. Int. Ed.* **59**, 16054-16060, (2020).

To Referee 2:

Question 1

In Figure S13, the excitation wavelength is 400 nm, but in the text the author says the $\lambda_{\text{ex}} = 390$ nm. So, which one is correct?

Response to question 1 of referee 2

Thanks for the careful reviewing. We have checked the original data to confirm the excitation wavelength of Supplementary Fig. 18 (Figure S13). All time-resolved decay curves of PzPh&TPA were measured under 400 nm excitation. Besides, we have repeated the measurement to ensure the accuracy. These two figures were listed below (Fig. R10) and the error between the two measurements was within permission. Therefore, the “PzPh&TPA was measured under 390 nm excitation” was replaced with “PzPh&TPA was measured under 400 nm excitation”.

Figure R10. Time-resolved decay curves of PzPh&TPA under 400 nm excitation at room temperature (green: original data; blue: repeated data).

Question 2

Line 169, the author describes the (lifetime) excitation dependence of PzPh&TPA from 390 to 365 nm, but there is no PzPh&TPA spectrum in Figure S9. Is the author describing the PzPh&BP?

Response to question 2 of referee 2

Thanks a lot for correcting our mistake. The excitation dependent decay curves of PzPh&TPA should be in

Supplementary Fig. 18. The mislabeling was corrected in the paper. The careful reviewing really improves our paper.

Question 3

Again, in Table S1, the excitation wavelength of PzPh&TPA has changed to 365/390 nm, which is inconsistent with the 400 nm stated in the text. Please revise it! Similar errors appear repeatedly. If these data are all in a muddle, the author had better be able to retest to ensure the reliability of the results.

Response to question 3 of referee 2

The reviewer helpfully corrected our mistake. We have confirmed that PzPh&TPA was measured under 400 nm excitation. To further confirm the correction, we have repeated the measurement to ensure the accuracy of the data. In addition, all the data in Table S1 were retested to ensure the reliability. These original data and repeated data are all listed below as Table R5-R6 and Fig. R11-R16. By comparing the original data and repeated data, it is revealed that the errors between the two serials of measurements are within permission. Therefore, the reliability of results could be further guaranteed. We really appreciate the reviewer's precise reviewing to improve our paper.

Table R5. Photophysical properties of crystalline powders for PzPh and host/guest materials (original data).

Sample	λ_{ex} (nm)	Fluo.			Phos.				
		λ_{F} (nm)	Φ_{F} (%)	τ_{F} (ns)	λ_{P} (nm)	Φ_{P} (%)	τ_{P1} (ms)	τ_{P2} (ms)	τ_{P3} (ms)
PzPh	365	445; 472	2.3	3.5	534; 577	-	-	-	-
PzPh&BP	365	445	1.0	6.0(445 nm)	543	11.4	10.6 (0.8%)	3.9 (99.2%)	-
	280	445	-	-	543	-	10.2 (3.4%)	4.8 (96.6%)	-
PzPh&TPO	365	445; 477	1.6	3.7(445 nm)	547	2.1	10.3 (29.3%)	21.5 (70.7%)	-
	280	292; 445	-	-	547	-	10.6 (19.3%)	20.5 (72.4%)	257.0 (8.2%)
PzPh&TPSi	365	445; 472	1.4	4.5(445 nm)	549	2.2	11.9 (79.3%)	30.4 (20.7%)	-
	280	305; 451	-	-	450; 549	-	11.3 (32.6%)	31.6 (8.7%)	778.9 (58.8%)
PzPh&TP	365	445; 475	1.3	3.8(445 nm)	547	9.0	23.3 (89.8%)	43.1 (10.2%)	-
	280	445	-	-	547	-	22.7 (84.4%)	43.2 (15.6%)	-
PzPh&SF	365	472; 498	5.9	5.4(445 nm)	556	4.4	22.6 (56.9%)	157.1 (43.1%)	-
	280	341; 492	-	-	494; 556	-	22.5 (6.7%)	154.4 (5.2%)	836.2 (88.1%)
PzPh&TPA	400	426; 445	4.6	5.1 (445 nm)	555	9.2	25.4 (65.8%)	376.9 (34.2%)	-

	365	390	-	-	400; 555	-	25.0 (70.0%)	428.1 (20.8%)	83.3 (9.2%)
--	-----	-----	---	---	----------	---	--------------	---------------	-------------

Table R6. Photophysical properties of crystalline powders for PzPh and host/guest materials (repeated data).

Sample	λ_{ex} (nm)	Fluo.			Phos.				
		λ_F (nm)	Φ_F (%)	τ_F (ns)	λ_P (nm)	Φ_P (%)	τ_{P1} (ms)	τ_{P2} (ms)	τ_{P3} (ms)
PzPh	365	445; 468	2.1	3.7	533; 576	-	-	-	-
PzPh&BP	365	420; 445	0.9	3.7(445 nm)	543	9.4	10.7 (0.7%)	3.4 (99.3%)	-
	280	418; 445	-	-	543	-	10.3 (3.4%)	4.6 (96.6%)	-
PzPh&TPO	365	445; 476	1.1	3.8(445 nm)	547	1.3	10.6 (27.2%)	21.4 (72.8%)	-
	280	292; 445	-	-	547	-	10.5 (19.9%)	20.6 (71.9%)	257.2 (8.2%)
PzPh&TPSi	365	445; 472	0.9	3.7(445 nm)	550	2.4	12.3 (80.2%)	32.5 (19.8%)	-
	280	303; 447	-	-	450; 550	-	11.1 (34.6%)	33.6 (7.5%)	785.2 (57.9%)
PzPh&TP	365	445; 471	1.3	3.0(445 nm)	548	8.9	23.8 (91.4%)	43.9 (8.6%)	-
	340	445	-	-	548	-	22.7 (84.4%)	43.2 (15.6%)	-
PzPh&SF	365	471; 498	5.7	5.6(445 nm)	556	4.4	22.6 (51.4%)	157.1 (48.6%)	-
	280	333; 493	-	-	493; 556	-	23.6 (4.0%)	158.1 (9.5%)	825.8 (86.5%)
PzPh&TPA	400	445; 477	9.3	3.0(445 nm)	555	6.9	25.4 (65.8%)	376.9 (34.2%)	-
	365	390	-	-	394; 555	-	25.4 (71.2%)	419.7 (20.8%)	91.1 (8.0%)

Figure R11. Photoluminescence spectra of PzPh and the mH/G UOP system under 365 nm excitation at room temperature (PzPh&TPA was measured under 400 nm excitation).

Figure R12. Photoluminescence spectra of PzPh and the mH/G UOP system under 280 nm excitation at room temperature (PzPh&TPA was measured under 365 nm excitation).

Figure R13. Delayed spectra of PzPh and the mH/G UOP system after the stoppage of 365 nm excitation at room temperature (PzPh&TPA was measured after the stoppage of 400 nm excitation).

Figure R14. Time-resolved decay curves of PzPh and the mH/G UOP system at 445 nm under 365 nm excitation at room temperature (PzPh&TPA was measured under 400 nm excitation).

Figure R15. Time-resolved decay curves of PzPh and the mH/G UOP system at 543 ~ 556 nm under 365 nm excitation at room temperature.

Figure R16. Time-resolved decay curves of PzPh and the mH/G UOP system at 543 ~ 556 nm under 280 nm excitation at room temperature (PzPh&TPA was measured under 400 nm excitation).

Question 4

In Figure 3b, when the $\lambda_{ex} = 280$ nm, the author obtains two lifetimes marked as " τ_2 ". Judging from the Table S1, it should be " $\tau_3 = 778.9$ ms".

Response to the question 4 of referee 2

We would like to thank the reviewer for the careful reviewing and correcting our mistake. The second τ_2 in Fig. 3b should be τ_3 . We have made the change to τ_3 , and double-checked other annotations in all the figures to avoid similar mistake.

Figure R1 (Fig. 3b). Time-resolved decay curves of PzPh&TPSi under 365 nm and 280 nm

excitation (inset: Photographs taken under a 365 or 254 nm UV lamp on and off).

Question 5

Line 176-179, the author described that the host TPSi and PzPh&TPSi have similar phosphorescence excitation spectra. Can the energy transfer of the doping system be fully proved by the excitation spectra alone? As far as I know, the possibility of energy transfer needs to be evaluated by the degree of overlap of the absorption spectra of the host and guest materials (because it involves triplet excitons, Figure 3e). In any case, further experiments are necessary to ensure that the energy transfer process proposed in the article is reliable.

Response to question 5 of referee 2

Thanks a lot for the reviewer's good question and helpful suggestion, the mechanism of energy transfer between TPSi and PzPh was not explained clearly in the previous manuscript. According to literatures, the triplet-exciton involved energy transfers mainly occurred by the short-range Dexter mechanism (< 1 nm).[43-46] It also requires an integral overlap between the emission spectrum of the donor and the normalized absorption spectrum of the acceptor. Follow the reviewer's suggestion, normalized absorption, phosphorescence excitation and phosphorescence emission spectra of the host and guest species were conducted, as shown in Fig. R17-R18 below. PzPh exhibited a broad absorption band with the maximum of 370 nm in crystalline state. While TPSi gave a narrow absorption band with the maximum of 269 nm, indicating a larger energy gap compared to the absorption band of PzPh. This provided evidence for the efficient energy transfer from the donor (TPSi) to acceptor (PzPh). In addition, the phosphorescence excitation spectrum of PzPh (at phosphorescence peak 534 nm) was overlapped with the phosphorescence emission spectrum of TPSi in a certain extent. It further demonstrated the possibility of Dexter energy transfer between the host and guest species. Dexter energy transfer was sensitive to the concentration of guest material according to previous literatures [45, 46]. To further confirm the Dexter energy transfer process, concentration-dependent emission studies for the PzPh&TPSi were performed (Fig. R19 and R20). As shown in Fig. R19, the ratio of phosphorescence bands was significantly enhanced as the concentration of PzPh increased. Correspondingly, the time-resolved decay curves at 549 nm decreased, indicating that the mH/G UOP system (PzPh&TPSi) showed a higher energy transfer efficiency at increased doping ratio of PzPh. These results further demonstrated the Dexter energy transfer mechanism for the mH/G UOP system.

Following the reviewer's suggestions, the following sentences have been added on page 12 (Line 191-204) "In the mH/G UOP system, triplet-exciton involved energy transfers were proposed as the short-range Dexter mechanism (< 1 nm) according to relative literatures⁴³⁻⁴⁶. To illustrate the energy transfer process in PzPh&TPSi, the absorption, phosphorescence excitation and emission spectra of the PzPh and TPSi were performed in Supplementary Fig. 21 and 22. PzPh exhibited a broad absorption band with the maximum of 370 nm in crystalline state. TPSi showed a narrow absorption band with the maximum of 269 nm, indicating a larger energy gap. In addition, the phosphorescence excitation spectrum of PzPh (at phosphorescence peak 534 nm) was overlapped with the phosphorescence emission spectrum of TPSi in a certain extent. These results demonstrated the possibility of Dexter energy transfer between the host and guest species. Dexter energy transfer was sensitive to the concentration of guest material according to previous literatures^{45,46}. As shown in Supplementary Fig. 23 and 24, the relative phosphorescence intensity of PzPh&TPSi (at 549 nm) showed an obvious enhancement and the time-resolved decay curves were shortened as the concentration of PzPh increased. These results indicated a higher energy transfer efficiency with the increasing PzPh doping ratio, and further demonstrated the Dexter energy mechanism for the mH/G UOP system."

Figure R17 (Supplementary Figure 21). Normalized absorption spectra of the crystalline powders for PzPh and TPSi.

Figure R18 (Supplementary Figure 22). Phosphorescence spectra of PzPh and TPSi and the excitation spectrum of PzPh at phosphorescence peak 543 nm taken at 77K.

Figure R19 (Supplementary Figure 23). Photoluminescence spectra of PzPh&TPSi under 280 nm excitation with different concentration (to facilitate comparison, the spectra were normalized at 304 nm).

Figure R20 (Supplementary Figure 24). Time-resolved decay curves of PzPh&TPSi at 549 nm under 280 nm excitation with different concentration.

43. Skourtis, S. S., Liu, C., Antoniou, P., Virshup, A. M. & Beratan, D. N. Dexter energy transfer pathways. *Proc. Natl. Acad. Sci. U. S. A.* **113**, 8115-8120, (2016).
44. Sun, M. J. *et al.* In situ visualization of assembly and photonic signal processing in a triplet light-harvesting nanosystem. *J. Am. Chem. Soc.* **140**, 4269-4278, (2018).
45. Du, B., Fortin, D. & Harvey, P. D. Singlet and triplet energy transfers in tetra-(meso-truxene)zinc(II)- and tetra-(meso-tritruene)zinc(II) porphyrin and porphyrin-free base dendrimers. *Inorg. Chem.* **50**, 11493-11505, (2011).
46. Wang, J.-X. *et al.* Organic composite crystal with persistent room-temperature luminescence above 650 nm by combining triplet-triplet energy transfer with thermally activated delayed fluorescence. *CCS Chem.* **2**, 1391-1398, (2020).

Question 6

It is recommended to add an annotation in Figure 3d: PzPh&TPSi. At the same time, the Rayleigh scattering is so strong that it overwhelms the fluorescence. Removing the Rayleigh scattering signal may be an effective option.

Response to question 6 of referee 2

Following the reviewer's helpful suggestion, an annotation PzPh&TPSi was added in Fig. 3d. In addition, we try to remove the Rayleigh scattering in Fig. 3d. Rayleigh scattering could be effectively eliminated with two methods: (1) using a suitable optical filter; (2) Delaunay triangulation method. For 3D excitation-emission mapping studies, it is quite difficult to choose a suitable optical filter due to the wide-range excitation wavelengths. Therefore, we adopt the Delaunay triangulation method to eliminate the Rayleigh scattering according to previous literatures [40-42]. The revised Fig. 3d in the paper was listed as the following. In addition, the legend of Fig. 3d was changed to "Excitation-emission mapping of PzPh&TPSi under ambient conditions (top: excitation-steady state emission mapping, the Rayleigh scattering was removed by Delaunay triangulation method⁴⁰⁻⁴²; bottom: excitation-delayed emission mapping; side: phosphorescence spectra taken under 365 nm (red line) and 280 nm (green line) excitation)". Similar Supplementary Fig. 36-40 were also revised as shown in Fig. R22-R26.

Figure R21 (Fig. 3d). Excitation-emission mapping of PzPh&TPSi under ambient conditions (top: excitation-steady state emission mapping, the Rayleigh scattering was removed by Delaunay triangulation method⁴⁰⁻⁴²; bottom: excitation-delayed emission mapping; side: phosphorescence spectra taken under 365 nm (red line) and 280 nm (green line) excitation).

40. Zepp, R. G., Sheldon, W. M. & Moran, M. A. Dissolved organic fluorophores in southeastern US coastal waters: correction method for eliminating Rayleigh and Raman scattering peaks in excitation-emission matrices. *Mar. Chem.* **89**, 15-36, (2004).

41. McKnight, D. M. *et al.* Spectrofluorometric characterization of dissolved organic matter for indication of precursor organic material and aromaticity. *Limnol. Oceanogr.* **46**, 38-48, (2001).
42. Fink, M., Haurert, J., Spoerhase, J. & Wolff, A. Selecting the aspect ratio of a scatter plot based on its Delaunay Triangulation. *IEEE T. Vis. Comput. Gr.* **19**, 2326-2335, (2013).

Figure R22 (Supplementary Figure 36). Excitation-emission mapping of PzPh&BP in ambient conditions (the Rayleigh scattering was removed by Delaunay triangulation method).

Figure R23 (Supplementary Figure 37). Excitation-emission mapping of PzPh&TPO in ambient conditions (the Rayleigh scattering was removed by Delaunay triangulation method).

Figure R24 (Supplementary Figure 38). Excitation-emission mapping of PzPh&TP in ambient conditions (the Rayleigh scattering was removed by Delaunay triangulation method).

Figure R25 (Supplementary Figure 39). Excitation-emission mapping of PzPh&SF in ambient conditions (the Rayleigh scattering was removed by Delaunay triangulation method).

Figure R26 (Supplementary Figure 40). Excitation-emission mapping of PzPh&TPA in ambient conditions (the Rayleigh scattering was removed by Delaunay triangulation method).

Question 7

"Wide-range" and "Lifetime-tunable" appear in the title of the manuscript. Reading through the full text, we find:

- 1) **This tunability is achieved by replacing the host molecule, and these host materials correspond to different phosphorescence lifetimes. Different UOP lifetimes are intrinsic properties of the host materials, not unique to the host-guest doping systems. Therefore, is it really appropriate to put "Lifetime-tunable" in the title? It's worth revisiting!**

Response to question 7(1) of referee 2

We appreciate the nice question from the reviewer. Our previous statement for the UOP lifetime tuning strategy is ambiguous. We have summarized the lifetimes and emission maxima of the mH/G UOP materials and their relative host matrices. The emission wavelengths of all the mH/G UOP system were consistent with the phosphorescence band of the guest molecule and quite different from the various host materials. By comparing the phosphorescence lifetimes of the mH/G UOP system and their relative host matrices, no obvious relationship could be summarized (Supplementary Table 1 and 2). On the contrary, the phosphorescence lifetimes of the mH/G UOP system were in perfect line with ΔE_{ScTh} (the energy gap

between the lowest singlet state of the guest emitter (S_{1G}) and the lowest triplet state of host matrices (T_{1H}) of the host matrices (Fig. R27 and Table R7). By enhancing the $\Delta E_{S_0T_1}$, a slower ISC process occurred between the host and guest species, and the phosphorescence lifetimes of the mH/G UOP system could be prolonged. Therefore, the wide-range lifetime-tunable properties of the mH/G UOP system are realized by tuning the lowest triplet states other than phosphorescence lifetimes of the host matrices. In summary: (1) The UOPs from the mH/G system originated from the triplet state of the guest molecule PzPh other than the host matrices; (2) The wide-range lifetime-tunable properties of the mH/G UOP system are realized by manipulating the T_{1H} and controlling the ISC processes other than changing the phosphorescence lifetimes of the host matrices.

To improve the paper and emphasize the UOP lifetime tunable strategy, a paragraph was added in page 8-9 (Line 145-153) as following: “By comparing the phosphorescence lifetimes of the mH/G UOP system and their relative host matrices, no obvious relationship could be summarized (Supplementary Table 1 and 2). Thus, the wide-range lifetime-tunable properties were not achieved by changing the phosphorescence lifetimes of the host matrices. On the contrary, the phosphorescence lifetimes of the mH/G UOP system were perfectly in line with $\Delta E_{S_0T_1}$ as shown in Supplementary Table 1 and 2. Therefore, it should be noted that the tunability of the mH/G UOP system was achieved by manipulating the T_{1H} of the host matrices and controlling the ISC processes.” This helpful suggestion could avoid the misunderstanding of the paper.

Figure R27 (Fig. 2b). Decay curves at the phosphorescence emission band of 543 nm ~ 556 nm under 365 nm excitation at room temperature.

Table R7. Photophysical properties of crystalline powders for the mH/G UOP materials.

	PzPh&BP	PzPh&TPO	PzPh&TPSi	PzPh&TP	PzPh&SF	PzPh&TPA
^a τ_{P1} (ms)	10.6 (0.8%)	10.3 (29.3%)	11.9 (79.3%)	23.3 (89.8%)	22.6 (56.9%)	25.4 (65.8%)
^a τ_{P2} (ms)	3.9 (99.2%)	21.5 (70.7%)	30.4 (20.7%)	43.1 (10.2%)	157.1 (43.1%)	376.9 (34.2%)
^b E_{SG1} (eV)	3.20	3.20	3.20	3.20	3.20	3.20
^b E_{TH1} (eV)	3.19	3.10	3.05	2.90	2.83	2.66
$\Delta E_{S_{0Tn}}$ (eV)	0.01	0.10	0.15	0.30	0.37	0.54

* a, Phosphorescence lifetimes of the mH/G UOP system were measured under 365 nm excitation at room temperature (PzPh&TPA was measured under 400 nm);

b, E_{SG1} and E_{TH1} were calculated according to the Tangent line of fluorescence spectrum of PzPh and phosphorescence spectra of the host materials (Supplementary Figure 9).

2) The author should provide afterglow photos or videos of the host materials to facilitate readers to compare the luminescence properties before and after doping. From the data presented so far, the biggest achievement of the doping systems is only to ensure the "uniformity" of the UOP color.

Response to question 7(2) of referee

Thanks for the good suggestion from the reviewer. The reviewer reminded us to clearly compare the afterglow performances of the mH/G UOP system with relative host matrices. Following the useful suggestion, we have added the afterglow photos of host and the mH/G UOP materials in Fig. R28 and R29 under different excitations. It was observed that there were no afterglows for the host materials after the stoppage of 365 nm excitation. On the contrary, all the mH/G UOP materials showed an analogical bright-yellow afterglow with continuously tunable lifetimes which gradually increased from 3.9 ms to 376.9 ms (Fig. R27 (Fig. 2b)). Under 254 nm excitation, phosphorescence emission could be detected for the host species except for TP. The mH/G UOP materials exhibited obvious differences in afterglow colors, phosphorescent spectral structures and decay lifetimes compared with their relative host species (detail statement see *Response to question 7(3)* below).

To further illustrate the different afterglow properties between the mH/G UOP system and the host materials. Supplementary Fig. 19 and 20 were added in the supporting information. In addition, sentences of "Different afterglows between the mH/G UOP system and the host materials (Supplementary Fig. 19) further demonstrated the different phosphorescent originations for the mH/G UOP system and the host materials." were added in page 9 (Line 148-150). We would like to show our great thanks to the reviewer for polishing our paper.

Figure R28 (Supplementary Figure 19). Photographs of host and the mH/G UOP materials at the conditions of "365 nm UV on" and "0.01 s UV off".

Figure R29 (Supplementary Figure 20). Photographs of host and the mH/G UOP materials at the conditions of "254 nm UV on" and "0.01 s UV off".

- 3) We note that the author had pointed out the possibility of high-energy excitation, however, we cannot evaluate its correctness before seeing the phosphorescence spectra of the hosts. So, the UOP spectra of the host molecules under different excitation should be supplemented. By comparing these spectra, the author needs to explain the origin of the excitation dependence of these materials. As we all know, if the guest and the host form an excimer, or there is a significant charge transfer state, it may lead to multi-color luminescence. In addition, if excitation dependence is the nature of the hosts, then the author should not emphasize this property again in the manuscript.

Response to question 7(3) of referee 2

Thanks for providing us with a helpful question. Our expression of the excitation-dependent UOP properties is ambiguous. To clearly illustrate the excitation-dependent properties of the mH/G UOP system. Supplemental experiments were performed and explained as follows:

According to the reviewer's suggestion, the UOP spectra of the hosts and the mH/G UOP materials under different excitation wavelengths (280 nm, 365 nm) were supplemented as shown in Fig. R30 to R35. Under these excitation wavelengths, no phosphorescence could be detected for the host of TP and phosphorescence of TPA was very weak. For other host materials, obvious phosphorescence emission could be detected under 280 nm excitation. By comparing the phosphorescence spectra of mH/G UOP materials and their relative host materials, obvious differences existed in phosphorescence maxima, spectral structures and decay lifetimes (Fig. 3a, Supplementary Fig. 12). These results indicated the different phosphorescence originations for the hosts and mH/G UOP system. Thus, the UOP bands for mH/G system were mainly ascribed to the phosphorescence from the guest molecules under the 365 nm excitation. Under 280 nm excitation, PzPh&BP, PzPh&TPO and PzPh&TPA ($\lambda_{\text{ex}} = 365 \text{ nm}$) showed the same phosphorescence spectra to those under 365 nm excitation (400 nm for PzPh&TPA), while two UOP bands observed in PzPh&TPSi and PzPh&SF were quite different from those under 365 nm excitation. By further comparing the phosphorescence spectra of PzPh&TPSi and PzPh&SF with their individual host materials (Fig. R36 and R37), it was observed that the phosphorescence bands of PzPh&TPSi and PzPh&SF at 280 nm were the superposition of the phosphorescence bands of the host matrices and the relative mH/G system under 365 nm excitation. These results showed that the UOP bands of the mH/G materials (excited at 280 nm) originated from the phosphorescence of the guest materials (main part) and the host materials (a small part). No obvious excimer during the luminescent processes is detected.

The reviewer pointed out our ambiguous expression of "excitation-dependent UOP" in the paper. Our previous statements were misleading that the excitation-dependent UOP mainly changed emission color for the mH/G system. Actually, the "excitation-dependent UOP" in the paper mainly described the great enhancement of phosphorescence lifetimes for some mH/G UOP materials under high-energy excitation (280 nm). In addition, it should be noted that the excitation-dependent lifetime tuning properties are aimed at the phosphorescence bands of guest molecules other than the host species. And the large-magnitude lifetime enhancing for some mH/G materials under higher-energy UV excitation was mainly due to multi-ISC pathways to form long-lived triplet excitons (as described in Figure 3e in the paper). Therefore, the excitation dependence of the mH/G UOP system was not related to the phosphorescence of the hosts.

To illustrate the “excitation-dependent UOP” clearer, sentences were added in page 12 (Line 217-222) as following: “The UOP spectra of the hosts and the mH/G UOP materials under different excitation wavelengths (280 nm, 365 nm) were shown in Supplementary Figure 30-35. For other host materials except for TP and TPA, obvious phosphorescence bands could be detected under higher-energy excitation. In the phosphorescence spectra, mH/G UOP materials and their relative host materials showed obvious differences in emission maxima, spectral structures and decay lifetimes (Fig. 3a, Supplementary Fig. 12).”.

To avoid misleading of the “excitation-dependent UOP” we should not emphasize the afterglow color changing properties too much. The original paragraph of “Noteworthy, minority of triplet excitons of the host matrices participated in phosphorescence through radiation transition under high-energy excitation. This led to the broadening of phosphorescence band in the doping systems, which was finally caused change of afterglow color (Fig. 3b, and Supplementary Fig. 32 and 34). This property further enriches the variability of excitation-dependent UOP in the multi-host/guest doping system.” was changed to “In addition, under high-energy excitation (280 nm), some triplet excitons of the host matrices contributed to phosphorescence through radiation transition. Thus, the UOP consisted of the phosphorescence from the mH/G system and the host species (a small part) were slightly broadened. Afterglow color changes might be observed (Fig. 3b, Supplementary Fig. 32 and 34). in page 12 (Line 222-225).

In addition, a sentence “Excitation-dependent UOP properties (showing great enhanced lifetimes for some mH/G materials under high-energy excitation) could further enrich the variability of the mH/G UOP system.” was added at the end of the paragraph (Page 13, (Line 225-227)).

Figure R30 (Supplementary Figure 30). Normalized delayed emission spectra of PzPh&BP and BP after the stoppage of different excitation wavelengths (virtual gating from 1 ms to 5 ms).

Figure R31 (Supplementary Figure 31). Normalized delayed emission spectra of PzPh&TPO and TPO after the stoppage of different excitation wavelengths (virtual gating from 1 ms to 5 ms).

Figure R32 (Supplementary Figure 32). Normalized delayed emission spectra of PzPh&TPSi and TPSi after the stoppage of different excitation wavelengths (virtual gating from 1 ms to 5 ms).

Figure R33 (Supplementary Figure 33). Normalized delayed emission spectra of PzPh&TP and TP after the stoppage of different excitation wavelengths (virtual gating from 1 ms to 5 ms).

Figure R34 (Supplementary Figure 34). Normalized delayed emission spectra of PzPh&SF and SF after the stoppage of different excitation wavelengths (virtual gating from 1 ms to 5 ms).

Figure R35 (Supplementary Figure 35). Normalized delayed emission spectra of PzPh&TPA and TPA after the stoppage of different excitation wavelengths (virtual gating from 1 ms to 5 ms).

Figure R36. delayed emission spectra of PzPh&TPSi and TPSi after the stoppage of UV excitation (virtual gating from 1 ms to 5 ms) and the corresponding divided spectrum.

Figure R37. delayed emission spectra of PzPh&SF and SF after the stoppage of UV excitation (virtual gating from 1 ms to 5 ms) and the corresponding divided spectrum.

Reviewer #1 (Remarks to the Author):

The points raised by the reviewers have been fully addressed. The already high level of the manuscript has certainly benefit from the additions/corrections. the manuscript can be published in the present form.

Reviewer #2 (Remarks to the Author):

The authors have responded all the questions properly and adequately, and the revised manuscript can be accepted.